# Health Hazard Associated with the Presence of *Clostridium* Bacteria in Food Products

**DOI:** 10.3390/foods13162578

**Published:** 2024-08-18

**Authors:** Agnieszka Bilska, Krystian Wochna, Małgorzata Habiera, Katarzyna Serwańska-Leja

**Affiliations:** 1Department of Food and Nutrition, Poznan University of Physical Education, Krolowej Jadwigi 27/39, 61-871 Poznan, Poland; bilska@awf.poznan.pl; 2Department of Swimming and Water Lifesaving, Poznan University of Physical Education, Krolowej Jadwigi 27/39, 61-871 Poznan, Poland; kwochna@awf.poznan.pl (K.W.);; 3Department of Sports Dietetics, Poznan University of Physical Education, 61-871 Poznan, Poland; 4Department of Animal Anatomy, Faculty of Veterinary Medicine and Animal Sciences, Poznan University of Life Sciences, Wojska Polskiego 71c, 60-625 Poznan, Poland

**Keywords:** microbiological food safety, *Clostridium* spp., *C*. *botulinum*, food pathogen, industrial application of *Clostridium* spp.

## Abstract

*Clostridium* bacteria were already known to Hippocrates many years before Christ. The name of the *Clostridium* species is owed to the Polish microbiologist, Adam Prażmowski. It is now known that these *Clostridium* bacteria are widespread in the natural environment, and their presence in food products is a threat to human health and life. According to European Food Safety Authority (EFSA) reports, every year, there are poisonings or deaths due to ingestion of bacterial toxins, including those of the *Clostridium* spp. The strengthening of consumer health awareness has increased interest in consuming products with minimal processing in recent years, which has led to a need to develop new techniques to ensure the safety of microbiological food, including elimination of bacteria from the *Clostridium* genera. On the other hand, the high biochemical activity of *Clostridium* bacteria allows them to be used in the chemical, pharmaceutical, and medical industries. Awareness of microbiological food safety is very important for our health. Unfortunately, in 2022, an increase in infections with *Clostridium* bacteria found in food was recorded. Knowledge about food contamination should thus be widely disseminated.

## 1. Introduction

*Clostridium* bacteria have been accompanying mankind for a very long time. The first report about them appeared around 430-370 B.C. thanks to Hippocrates. In his book, Epidemics III, he described a disease that was diagnosed as gas gangrene and was caused by the *Clostridium* histolyticum bacteria [1]. A description of another, now well-known, disease, tetanus, appeared in 1824 in Charles *Bell’s Essays on the Anatomy and Philosophy of Expression* [1]. Another groundbreaking event focusing on bacteria of the genus *Clostridium* was Louis Pasteur’s description in 1861 of a microorganism capable of growing in the absence of oxygen, which was a huge sensation at the time. Pasteur named this bacterium *Vibrion butyrique* because of the main product of its fermentation, butyrate, and also coined the term “anaerobic” to describe life without free oxygen. This microorganism was named *Clostridium butyricum* 20 years later by Polish microbiologist Adam Prażmowski. This name still functions today [2].

*Clostridium* bacteria are widely distributed in the natural environment (they occur, among others, in dust, soil, water, bottom sediments, and in the digestive tract of humans and animals) [3]. They colonize both plant and animal raw materials, which poses a threat to human health. The increase in the consumption of low-processed products in recent years has resulted in an increase in the number of epidemics of foodborne diseases caused, among others, by food poisoning by bacteria of the genus *Clostridium* [4]. Therefore, many scientific studies are currently devoted to methods of effective elimination of *Clostridium* from ready-made food products [5,6,7].

Although bacteria belonging to *Clostridium* spp. are mainly associated with pathogenicity, their significant biochemical activity makes them useful in many industries (e.g., for the production of carbon dioxide, hydrogen, acids, and organic solvents) [8,9].

The paper discusses some aspects of the presence of *Clostridium* bacteria in food, with particular emphasis on the risks to human health. An important aspect of the work is collecting statistical data on the number of outbreaks of food poisoning caused by bacterial toxins, including *Clostridium* spp. The paper collects and compiles European Food Safety Authority reports dating back to 2013.

## 2. General Characteristics of the Genus *Clostridium* spp.

Clostridial cells are gram-positive or gram-variable bacilli of various sizes. Under stress conditions, they produce oval or spherical spores. Bacteria belonging to this genus are anaerobes; they do not produce catalase [3,10] (see Figure 1). The optimum growth temperature for most species of this genus is between 30 and 40 °C, although thermophilic species are also known, for which the optimum growth temperature is 60 and 75 °C [11]. Among *Clostridium* bacteria, however, there are also those that do not show typical morphological features for this genus. These include *C. coccoides*, which forms round cells; *C. perfingens*, *C. leptum*, *C. barati*, and *C. spiroforme*, in which spores are difficult to detect or do not form; and *C. tertium*, *C. carnis*, *C. histolyticum*, *C. intestinalis*, and *C. bifermentans*, growing in the presence of oxygen [9,12,13].

### Pathogenicity

The genus *Clostridium* includes more than 100 species, of which approximately 35 are pathogenic exotoxin-producing species, including *C. botulinum*, *C. perfingens*, *C. tetani*, *C. difficile*, *C. barati*, *C. haemolyticum*, *C. novyi*, *C. septicum*, and *C. chauvoei* [3].

The best-known *Clostridium* pathogen is *C. botulinum*, which causes botulism. *C. botulinum* cells are straight, slightly hook-like, gram-positive (in young cultures), rod-shaped, anaerobic, 0.5–2.0 μm wide, and 1.6–2.2.0 μm long, with oval and last spores [14]. Food poisoning is caused by intoxication with *botulinum* neurotoxin—botulism—produced by these microorganisms in food products [15,16]. So far, eight *botulinum* toxins have been recognized: A, B, Cα, Cβ, D, E, F, and G. Poisoning in humans can be caused by toxins A, B, E, and, rarely, F. These toxins are proteins resistant to acids (also hydrochloric acid in gastric juice) and low temperature. They are destroyed at 80–90 °C. The initial symptoms of botulism occur after 24–48 h and include abdominal pain, nausea, vomiting. Then, there are symptoms from the nervous system, such as visual disturbances, swallowing difficulties, speech and breathing disorders, and drooling. Untreated botulism leads to death by suffocation or cardiac arrest. It has been shown that *botulinum* neurotoxins are also produced by *C. butyricum* (type E toxin) and *C. barati* (type F toxin), as well as *C. subterminale* and *C. hastiforme*, but diseases caused by these species are found sporadically [17,18,19,20,21]. Neurotoxins C and D produce diseases in fowls and mammals. The type G, known since 1970, has not been established as a reason behind sickness in humans or animals. The mechanisms of action and structure of all isomers of neurotoxins are analogous. Every toxigenic clostridium yields a peptide which is excited by proteases once microorganism lysis. The capability of this bacteria to cause sickness in humans is directly associated with the assembly of heat-resistant spores that survive maintenance approaches and kill nonsporulating organisms. The warmth resistance of spores fluctuates from kind to kind and, additionally, from strain to strain Though some strains do not survive at 80 °C, the spores of many strains would need temperatures on top of the boiling to confirm their destruction [14].

*C. perfringens* is a gram-positive and spore-forming anaerobic bacillus which ubiquitously resides in nature in animal microbiota, soil, and decaying plants, and also in marine sediments. Despite the fact that *C. perfringens* is an anaerobic bacterium, it can still survive in the presence of oxygen and under low concentrations of superoxide. Additionally, it has been noticed that *C. perfringens* can potentially survive in aerobic environments (such as surfaces in hospital wards) and can initiate disease course in aerophilic environments and in oxygen-exposed tissues (gas gangrene), which may facilitate bacterial host-to-host transmission [22]. *C. perfringens* has induced a large array of severe diseases over the centuries in the muscle, gut, and other organs or tissues, such as gas gangrene and foodborne or non-foodborne poisoning, leading to diarrhea, enterotoxaemia, necrotizing, and necrotic enteritis. Besides severe life-threatening complications and numerous mortalities, it is estimated that *C. perfringens*-induced diseases cause USD 0.2–1.7 billion yearly in human foodborne enteritis in the USA alone and USD 6 billion yearly in the poultry industry around the world (data from 2022). Successful induction of different diseases, in part, comes from *C. perfringens*’ ability to produce more than 20 pathogenic toxins and enzymes in various combinations [23]. The strains belonging to this species are divided into 7 types (A–G) depending on the type of toxin produced. Types A and C are most commonly responsible for human infections, among other gas gangrene. Other species of *Clostridium* may also participate in the development of gas gangrene, e.g., *C. novyi* and *C. septicum*. It is estimated that about 5–6% of *C. perfringens* strains (mainly type A) are capable of producing enterotoxins. Consumption of food containing such vegetative cells causes food poisoning manifesting in severe diarrhea and abdominal pain, usually lasting 8–24 h. In addition, enterotoxigenic strains are also responsible for the so-called occasional diarrhea and, in about 5–20% of cases, for post-antibiotic diarrhea. *C. perfringens* can also cause necrotizing enterocolitis. The elderly and immunocompromised patients are particularly susceptible to infections caused by *C*. *perfringens* [11,16,24,25,26,27]. It should be emphasized that *C. perfringens* is a widely occurring pathogen in nature, and moreover, unlike other anaerobic bacteria infecting limited animal hosts and their tissues, *C. perfringens* has a successful living spectrum from the muscles to the gut. Induced diseases show complex manifestations of rapid bacterial overgrowth, rapid gas accumulation, collateral inflammatory self-destruction, and, additionally, various toxin productions [23].

*C. difficile* is the leading cause of nosocomial diarrhea in the developed world. According to the Centers for Disease Control and Prevention, *C. difficile* is a major nosocomial pathogen with more than 220,000 infections, 13,000 deaths, and nearly USD 5 billion in annual treatment associated costs, which are predicted to increase in the future [28,29]. *C. difficile* infection in humans has not been proven to be transmitted directly from animals, food, or the environment. Several studies have found *C. difficile* in a variety of foods, including meat, raw milk, vegetables, and seafood, supporting the theory that spore-contaminated foods could be contributing to *C. difficile* exposure and transmission. The presence of *C. difficile* in animals and common PCR ribotypes between humans and animals helps in zoonotic transmission [30]. *C. difficile* has been isolated from many domestic and wild animals, including camels, horses, donkeys, dogs and cats, domestic fowl, seals, and snakes. However, reports of disease in wild species, including cases in a Kodiak bear, a rabbit, a penguin, and captive ostriches, are sporadic. This organism is a major cause of diarrhea and fatal necrotizing enterocolitis in foals and nosocomial diarrhea in adult horses [31]. The main virulence factors responsible for the onset of symptoms during *C. difficile* infection, including diarrhea and pseudomembranous colitis, are the monoglucosyltransferases Toxin A and Toxin B [32,33]. The A toxin is a strong cytotoxin and an enterotoxin that causes diarrhea, released by vegetative cells. These bacteria are also the cause of 25% of all diarrhea cases developing after an oral administration of antibiotics [11]. *C. difficile* has also been shown to be a pathogen in animals, e.g., pigs and poultry, as well as dogs, cats, and horses, but the source of this type of infection is still unknown. Little is known about possible routes of transmission of *C. difficile* from animals to humans. It is known that the consumption of meat from sick animals can lead to infection, but other possible transmission routes are still being analyzed [34,35].

According to the report of the European Food Safety Authority (EFSA), the number of outbreaks of food poisoning caused by bacterial toxins (including toxins of *Clostridium* spp., *Bacillus* spp., *Staphylococcus* spp.) has remained at a similar level between 2013 and 2022, namely, 848 on average per year (min. 527, max 1141) (Table 1). Thus, symptoms of food poisoning occur in almost 10,000 people, and several hundred hospitalizations and deaths are also recorded. With the exception of 2020, the number of food poisoning cases has remained at a similar level or at an increase. Among the analyzed 10 years, 2020 recorded the lowest number of outbreaks of food poisoning (527, including 41 caused by *Clostridium* spp). The share of *Clostridium* spp. bacterial toxins being the cause of outbreaks of food poisoning was the highest in 2013 (20.4%) and included 170 outbreaks (including 3530 cases, 66 hospitalizations, and 1 fatal case). Most cases were characterized by symptoms of mild intoxication. In 2021, 47 food poisoning outbreaks caused by *Clostridium* spp. toxins were observed (6.9% of the total number of food poisoning outbreaks caused by bacterial toxins), including 7 caused by *Clostridium botulinum* toxins and 40 by *Clostridium perfringens*. Food poisoning symptoms occurred in 802 patients, of whom 40 required hospitalization, and 4 fatal cases were recorded. Although the number of foodborne outbreaks involving bacterial toxins increased in 2021 (152 foodborne outbreaks more than in 2020), it was still lower on average than for the period 2017–2019 (242 fewer foodborne outbreaks; a 26.3% relative fall compared with 2017–2019). However, despite the relatively small share of *Clostridium* spp. toxins in outbreaks of food poisoning, it is this group that is the most common cause of death as a result of food poisoning with bacterial toxins (see Table 1). *Clostridium* spp. toxins were the cause of approximately 3% of all reported outbreaks of food poisoning in Europe [36,37,38,39,40,41,42,43,44,45,46].

In Poland, only two cases of strong symptoms of food poisoning with botulinum toxin were recorded in 2013, and two in 2015 (two cases of mild poisoning were reported in 2014). For comparison, in Denmark in 2013, there were 682 cases of severe food poisoning caused by bacteria belonging to *Clostridium* spp.; in Great Britain, 510; and in France, 482 (including 1 fatal case) [36].

In 2019, in the EFSA report, out of 10 pathogen/food pairs causing the highest number of strong-evidence outbreaks, there was one pair of *Clostridium perfringens*/meat and meat products responsible for 19 outbreaks, including France (5), Spain (4), Denmark (3), Italy (2), United Kingdom (2), Germany (1), Hungary (1), and Greece (1). However, among the 10 pathogen/food pairs causing the highest number of cases in strong-evidence outbreaks, there were *Clostridium perfringens*/meat and meat products—589 cases, including France (159), Spain (154), Denmark (74), Greece (58), United Kingdom (56), Italy (55), Hungary (21), and Germany (12)), and *Clostridium perfringens*/mixed food (507 cases, including Denmark (268), France (115), Portugal (60), Sweden (34), and United Kingdom (30) [42]. For comparison, in 2020, among the 10 pathogen/food pairs causing the highest number of strong-evidence outbreaks, there was one *Clostridium perfringens* toxins/mixed food pair responsible for 8 outbreaks, including France (2), Denmark (2), Finland (1), Germany (1), Italy (1), and Portugal (1). However, among the 10 pathogen/food pairs causing the highest number of cases in strong-evidence outbreaks, only *Clostridium perfringens* toxins/mixed food was found—292 cases, including Denmark (45), Finland (42), France (41), Germany (16), Italy (128), and Portugal (20) [43]. In 2021, the pair was the same, but there were fewer cases, 161 including Italy (69), France (39), Portugal (20), Germany (15), Finland (12), and Denmark (6) [35]. The 2021 EFSA report did not include *Clostridium* spp. pathogens in the top 10 pathogen/food pairs causing the highest number of strong-evidence outbreaks [44].

According to EFSA reports, in 2022, nine member states of the European Union (Belgium, Denmark, Finland, France, Italy, Portugal, Slovenia, Spain, and Sweden) reported FBOs caused by *Clostridium perfringens* toxins. This pathogen was associated with the largest mean outbreak size (52.7 cases). FBOs caused 1869 cases altogether, and the largest outbreak was reported by Portugal. *Clostridium perfringens* toxins was the causative agent involving the highest number of human cases, paired with “other or mixed red meat and products thereof” with961 cases, including Portugal (950), Finland (8), and France (3); “unspecified meat and meat products” (Spain—368 cases); and “other food” (Spain—266 cases) [45].

Eight member states of the European Union (including Poland) and four non-MSs reported more foodborne outbreaks in 2021 than in 2020, but fewer than in the pre-pandemic years of 2017–2019, on an average. Eleven member states of the European Union and Serbia reported fewer foodborne outbreaks in 2021 on average than in both 2020 and the period of 2017–2019. Taken together, these results suggest that the COVID-19 pandemic and the associated control measures continued to have a major impact in 2021 on the occurrence of foodborne outbreaks and their reporting in the European countries. In 2022, the reporting rate of FBOs caused by bacterial toxins was 0.25 per 100,000 population. This was a relative increase of 68.2% compared with the rate in 2021, owing mainly to the increased reporting of FBOs associated with Bacillus cereus toxins [45].

Table 1 shows the number of food poisonings caused by bacterial toxins, including *Clostridium* spp. toxins.

## 3. *Clostridium* Bacteria in Food Products

Microbial contamination of food raw materials is an important factor affecting food safety, as the presence of pathogenic bacteria can be a reason for food poisoning in humans.

Each year worldwide, unsafe food causes 600 million cases of foodborne diseases and 420,000 deaths. In fact, 30% of foodborne deaths occur among children under 5 years of age. The World Health Organization (WHO) estimates that 33 million years of healthy lives are lost due to eating unsafe food globally each year, and this number is likely an underestimation.

In recent years, there has been an increase in consumer interest in products with a minimum degree of processing, which are manufactured according to the technology of gentle processing of raw materials, conducive to preserving their natural features; unfortunately, this brings a high number of poisonings [47]. The exclusion of high-temperature methods during product preservation is obviously beneficial in terms of preserving its nutritional values, but it only reduces the number of microorganisms and does not eliminate them completely [47]. Low-processed products, including portioned meat, fermented milk, milk drinks, fish, fruit, and vegetables, may be contaminated with bacteria, yeasts, molds, and viruses, not only leading to unfavorable organoleptic changes in the product, but also presenting a direct threat to health and consumers’ lives. Soil microflora is particularly dangerous for human health, such as spore-forming bacteria of the genera *Clostridium*, *Bacillus*, *Staphylococcus*, *Escherichia*, and *Pseudomonas*. According to the literature, the main cause of food poisoning is the presence of Campylobacter spp., *Salmonella* spp., *Staphylococus* spp., *Escherichia* spp., *Pseudomonas* spp., and *Clostridium* spp. in ready-made food products [48]. *Clostridium* bacteria are particularly characteristic of several food groups. They form the dominant microflora of root and tuber vegetables. They are also found in meat, poultry, fish, and seafood, as well as in spices (including pepper, oregano, and cinnamon). In addition, hermetic packaging of these products increases the risk of microflora r proliferation [49,50].

Food poisoning is caused not only by pathogenic bacteria, but also by toxins produced by them. A particular problem is the presence of *botulinum* neurotoxins in food [20]. Botulinum neurotoxins, one of the most potent toxins among biological substances, which are produced during the germination of *C. botulinum* spores, are the direct cause of botulism and the symptoms of botulism in infants and adults. Neurotoxins designated as BoNT/A, BoNT/B, and BoNT/E cause as much as 99% of human diseases caused by bacterial toxins [16,51,52]. It has been shown that some strains of *C. botulinum* are able to grow and produce toxins at refrigeration temperatures [53]. They are also found in frozen convenience food—products of this type are most often subjected to incomplete thermal processing before freezing, which is not a procedure that allows for the destruction of *Clostridium* bacterial spores [54,55]. *C. botulinum* bacteria are characteristic of meat and fish because they occur in their digestive tracts [16]. Their presence is often found in canned vegetables and meat, smoked and cured meat, as well as in salted and smoked fish. Such contaminated food is characterized by the smell of rancid fat and the presence of gas. In canned food, the presence of *C. botulinum* causes bombage. It should be noted that a lethal dose of *botulinum* toxin for humans can be produced by a small amount of bacteria, with which sensory changes in food have not yet been observed [16,56].

In recent years, interest in products for vegetarians has also been growing. Although vegetarian sausages have not been linked to botulism, numerous outbreaks of the disease caused by canned vegetables suggest the frequent occurrence of *C*. *botulinum* spores in the raw material. Vegetarian sausages contain a limited amount of preservatives, and their shelf life may be several months. The safety of this product therefore depends mainly on heat treatment and cold storage. A major food safety concern is *C*. *botulinum* group II, which can grow and produce toxins at refrigerated temperatures. The authors of the study in [57] observed a high overall incidence of *C. botulinum* in samples of vegetarian sausages from various manufacturers. Strains of both groups I and II, as well as neurotoxin genes of types A, B, E, and F, were detected in the products. The highest number of cells was observed for *C. botulinum* group II in products with a remaining shelf life of 6 months at the time of purchase. Therefore, vacuum-packed vegetarian sausages often contain *C. botulinum* spores and may carry a high risk of *C. botulinum* growth and toxin production [27].

Bacteria of the species *C. perfringens*, thanks to their ability to survive in a variety of environments, are present not only in soil and sewage, but also in the digestive tracts of humans and animals. The carriers of this bacterium are often workers employed in the production or distribution of food. The source of *C. perfringens* infections is most often poultry, beef, and meat products, as well as spices, fish, and seafood [58,59]. Due to the fact that poisoning is caused by a relatively high number of bacteria, raw food or food that has undergone mild heat treatment and then stored at room temperature for several hours is most often responsible for infection [16,26,27,49].

*C. difficile* is still not officially recognized as a food pathogen, but scientific reports continue to provide new evidence of food spoilage caused by this species. This inconsistency is due to the inability of *C. difficile* to grow in a bile-salt-free environment. The presence of *C. difficile* bacteria has so far been confirmed in meat, fish, and seafood, as well as in root vegetables. These infections can originate from raw materials, such as raw vegetables, and fermented and smoked products. However, the possibility of cross-infection associated with the transmission of *C. difficile* by domestic animals should also be considered. The nature of *C. difficile* as a food pathogen still requires additional research [60,61].

Individual species of *Clostridium* show saccharolytic and proteolytic activity, which is why they can cause spoilage of a number of food products. The delay in the development of lactic acid bacteria in silage favors the proliferation of *C. butyricum* and *C. tyrobutyricum*, which are saccharolytic species, resistant to low pH. These bacteria cause the occurrence of a very pungent butyric acid odor in these products, making them unfit for consumption [62]. The chemical composition of canned vegetables, vegetable–meat, and meat products enables the development of *Clostridium* bacteria, such as *C*. *perfringens* and *C. sporogenes*, which cause the canned food to bomb. These bacteria can be responsible for the spoilage of milk and ripened cheeses. *Clostridium* bacilli are also able to break down the starch contained in flour and potato products—this takes place by fermentation when the cold chain is not maintained. The significant biochemical activity of bacteria of the *Clostridium* genus makes them a factor responsible for the spoilage of many commonly consumed products stored without access to oxygen, which is a burdensome issue for both the consumer and the food manufacturer [49,62,63,64].

According to EFSA reports in 2013, the main categories of food that were causes of food poisoning due to bacteria of the *Clostridium* genus were beef and related products, as well as mixed food. In the same year, an outbreak of food poisoning caused by enterotoxigenic strains of *C. perfringens* was reported in Belgium, involving 70 cases. These bacteria were detected in the rest of the stew (at the level of 6 log CFU/g). An epidemiological investigation showed that once the dish was prepared, it was kept refrigerated for 24 h and then reheated before consumption. Insufficient cooling of the goulash before placing it in the refrigerator created excellent conditions for the development of *C. perfringens* [36].

In 2014, in addition to the food categories listed in 2013, preserved food and pork were also listed as the main sources of pathogenic bacteria of the *Clostridium* genus (other foods and beef and derived products). In 2014, two outbreaks of severe poisoning caused by *C. perfringens* toxins were recorded in Denmark, which involved 391 cases (11.9% of all cases caused by *Clostridium* spp. toxins) and were caused by the consumption of mixed meals (different food). The reason for the development of bacteria was insufficient cooling of the food the day before serving [37].

The EFSA report for 2015 already includes food categories that have resulted in major outbreaks of food poisoning caused by *C. botulinum* (meat products, pork and derived products, smoked ham, preserved food, mixed food, and cereal products) and *C*. *perfringens* (meat and meat products—mainly beef, pork and poultry meat, and mixed food, but also mutton meat and vegetable juices) [38].

In recent years, in addition to the food categories listed by EFSA that cause the most outbreaks of food poisoning caused by *C. botulinum* and *C. perfringens*, meat substitutes for vegetarians (vegetarian sausages, vegetarian home-canned pate) have also been mentioned [27,65].

In 2019, in the EFSA report, among the 10 pathogen/food carrier pairs causing the highest number of deaths in outbreaks with strong evidence, there were *C. perfringens*/food of non-animal origin (France, two deaths), *C*. *botulinum*/other foods (Poland, one death). and *C. perfringens*/meat and meat products (Italy, one death) [42]. The 2020 EFSA report listed only three pathogen/food pairs causing the highest number of deaths in outbreaks with strong evidence, and *Clostridium* spp. was not among them [43].

For comparison, second on the list of the top seven pathogen/food vehicle pairs causing the highest number of deaths in strong-evidence outbreak in 2021 was the pair *Clostridium perfringens* toxins/pig meat and products thereof (France, 3 deaths) [44].

The problem of food poisoning caused by *Clostridium* bacteria affects the whole world. The incidence of gastroenteritis outbreaks in Singapore was analyzed from 2018 to 2021. Among foodborne outbreaks (n = 121), about 42.1% of outbreaks involved food prepared by caterers, 14.9% by restaurants, and 12.4% by in-house kitchens. *C. perfringens* and *Salmonella* were the most common pathogens causing foodborne outbreaks [66].

## 4. Other Aspects of the Presence of Bacteria of the Genus *Clostridium* in the Environment

Non-pathogenic bacteria *Clostridium* spp., due to the production of numerous extracellular enzymes, are widely used in industry. They can be used, among others, in the biotechnological production of butyric acid, some solvents (e.g., butanol, acetone and isopropanol), diols (e.g., 1,3-propanediol, 2,3-butanediol) [67], ammonia, and hydrogen [8]. There is a possibility of using products resulting from the action of bacteria of the *Clostridium* genus in the production of biofuels [35,68]. In the intestines of animals and humans, *Clostridium* species mostly utilize indigestible polysaccharide. And most of the metabolites they produce bring many benefits to gut health, such as short-chain fatty acids, bile acids, and bioactive proteins [69].

They are also used in medicine. Over the past few years, botulinum neurotoxin has been transformed from a cause of life-threatening affliction to a medical therapy. In 1978, Dr. Alan Scott was the first to use botulinum neurotoxin in humans for treatment of strabismus. Nowadays, after elucidating the pharmacological mode of the botulinum toxin action, it has become possible to use it in a wide spectrum of health disorders. Botulinum toxin is effective in the treatment of some pain syndromes, e.g., it can selectively weaken painful muscles by interrupting the spasm pain cycle, and is well tolerated in the treatment of chronic pain disorders in which pharmacotherapy can have side effects (such as migraines, chronic lumbar pain, tension headaches, and myofascial pain). Additionally, injections of the botulinum toxin are among the latest means of therapy for the treatment of neurological diseases (spasticity, in particular cerebral palsy, Parkinson’s disease, and Tourette’s syndrome), gastroenterological diseases (achalasia), urological diseases (detrusorsphincter dyssynergia, detrusor instability, lower urinary tract dysfunction), ophthalmological (strabismus), or dermatological diseases (hyperhidrosis, facial flushing) [70].

In the cosmetic industry, the procedure performed by intramuscular injection of BoNT is aimed to reduce facial wrinkles. Facial wrinkles are formed primarily due to intensive work of leading muscles, wrinkling one’s brows and longitudinal muscle. Treatment of facial wrinkles involves injecting different doses of botulinum toxin into the muscle, which reduces facial muscle activity. A significant improvement in facial skin tension is observed in approximately 90% of patients [70].

Some *C. butyricum strains* are also considered to be beneficial for human health, and they represent approximately 10–20% of human fecal samples. This bacterium has been widely used as a probiotic in Asia (particularly in Japan, Korea, and China). For example, the C. butyricum MIYAIRI 588 isolated from human stool samples by Chikaji Miyairi in 1933 and in 1963 from soil samples has been used as a probiotic for decades. This strain is a probiotic commercially available in Japan and Korea, and is used for supporting the treatment of antimicrobial-associated diarrhea. Moreover, the mentioned strain was also authorized as a novel food ingredient by the European Parliament and the European Council. *C. butyricum* is able to produce short-chain fatty acids by fermenting undigested dietary fiber, especially butyrate and acetate. The literature data also indicate the beneficial effects of the application of *C. butyricum*, such as promoting faster animal growth and enhancing different immune functions as well as microecological balance. A preventive effect against *Esherichia coli* and *C. difficile* infections and its influence on the reduction in intestinal damage and permeability is also demonstrated [71].

*C. sporogenes* spores can be used in the treatment of cancer—these bacteria, during the colonization of cancer cells, produce proteases inside the tumor, leading to its degradation. Over the past decade, there has been considerable interest in exploiting the industrial potential of *C*. *botulinum* and *C. tetani*. Intensive research is conducted in order to examine the structure, physiology, and biochemistry of neurotoxins produced by these bacteria [35,52,70,72,73,74,75], as well as the use of a complex of *botulinum* toxins as therapeutic agents in the treatment of human diseases [76,77]. There are also reports on the production of bacteriocins by such bacteria as, among others, *C. sporogenes*, *C. butyricum*, *C. botulinum* [78], *C. perfingens*, *and C. acetobutylicum* [79,80]. In addition, pectinolytic *Clostridium* bacteria loosen the tissue structure of plants, facilitating the separation of cellulose fibers, which makes them useful in the initial cleaning of flax and hemp [11].

## 5. Conclusions

*Clostridium* bacteria are widely distributed in the natural environment and occur, among others, in soil, water, and human and animal feces. They colonize both plant and animal raw materials, which poses a threat to human health. According to European Food Safety Authority (EFSA) reports, every year, there are poisonings or deaths due to ingestion of bacterial toxins, including those of the *Clostridium* spp. However, it should not be forgotten that bacteria of the *Clostridium* genus also have a variety of positive properties, as mentioned in the paper. Therefore, their beneficial use and applications are likely to expand in the near future in industry, medicine, health care, science, and different branches of economy worldwide.

## Figures and Tables

**Figure 1 foods-13-02578-f001:**
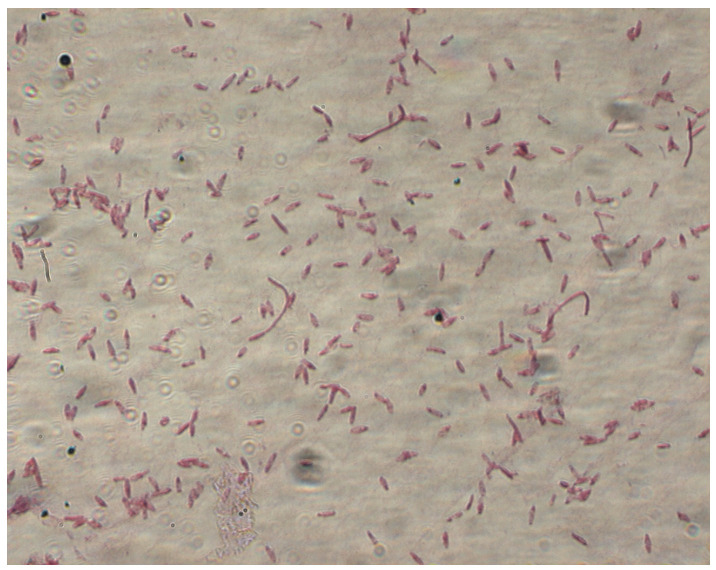
*Clostridium butyricum* (photo taken at the Department of Biotechnology and Food Microbiology at the University of Life Sciences in Poznań using an inverted fluorescence microscope( Zeiss, Axiovert 200)).

**Table 1 foods-13-02578-t001:** Number of food poisonings caused by bacterial toxins, including *Clostridium* spp. toxins.

	2013 EFSA [36]	2014 EFSA [37]	2015 EFSA [38]	2016 EFSA [39]	2017 EFSA [40]	2018 EFSA [41]	2019 EFSA [42]	2020 EFSA [43]	2021 EFSA [44]	2022 EFSA [45]
Number of outbreaks of food poisoning caused by bacterial toxins	834	840	849	848	818	950	997	527	679	1141
Number of cases (hospitalizations/fatalities)	9203 (452/1)	8610 (586/5)	8847 (497/3)	8967 (401/1)	8468 (583/7)	9726 (534/6)	10,555 (361/14)	4517 (182/6)	6378 (310/7)	13,902 (416/11)
Number of outbreaks of food poisoning caused by *Clostridium* spp. toxins	170 no data	160 including: *Clostridium botulinum*—9 *Clostridium perfringens*—124 Others *Clostridium* spp.—27	122 including: *Clostridium botulinum*—24 *Clostridium perfringens*—96 Others *Clostridium* spp.—2	*Clostridium botulinum*—18	*Clostridium botulinum*—9	86 incuding: *Clostridium botulinum*—15 *Clostridium perfringens*—71	82 incuding: *Clostridium botulinum*—7 *Clostridium perfringens*—75	41 including: *Clostridium botulinum*—9 *Clostridium perfringens*—32	47 including: *Clostridium botulinum*—7 *Clostridium perfringens*—40	62 including: *Clostridium botulinum*—7 *Clostridium perfringens*—55
Number of cases (hospitalizations/fatalities)	3530 (66/1) no data	3285 (65/3) no data	2074 (68/3) incuding: *Clostridium botulinum*—60 (43/0) *Clostridium perfringens*—2014 (25/3) Others *Clostridium* spp.—4 (no data)	*Clostridium botulinum*—49 (39/0)	*Clostridium botulinum*—26 (26/2)	1831 (53/4) incuding: *Clostridium botulinum*—48 (35/2) *Clostridium perfringens*—1783 (18/2)	2443 (42/4) incuding: *Clostridium botulinum*—17 (15/1) *Clostridium perfringens*—2426 (27/3)	716 (44/2) incuding: *Clostridium botulinum*—34 (34/0) *Clostridium perfringens*—682 (10/2)	802 (40/4) incuding: *Clostridium botulinum*—24 (15/0) *Clostridium perfringens*—778 (25/4)	2917 (21/3) incuding: *Clostridium botulinum*—20 (10/0) *Clostridium perfringens*—2897 (11/3)

## Data Availability

No new data were created or analyzed in this study. Data sharing is not applicable to this article.

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
