# Peer review of "Health Hazard Associated with the Presence of Clostridium Bacteria in Food Products"

_foods, 2024, doi:10.3390/foods13162578_

Round 1
Reviewer 1 Report
Comments and Suggestions for Authors
Bilska et al submitted to Food journal a review on associated health hazards linked with the presence of Clostridium in food products.
The overall review touches a hot topic in relation to the presence of foodborne bacteria, such as Clostridium, in foods. However, the review lacks proper structure with the title of this review not truly reflected in the whole review paper.
Instead, it summaries the findings in relation to foodborne outbreaks and incidence of cases from EFSA reports, which EFSA reports themselves are able to provide these details in a summarized format. Perhaps, a more comprehensive data analysis by linking comparisons between years of associated outbreaks, different biases to take into consideration, or other research studies beside EFSA reports would have been more appropriate to include and discuss in the review.
The introduction would be more suitable for a book chapter rather than journal.
The figure is not visible.
Comments on the Quality of English LanguageThe quality of English must be revised throughout the manuscript.
Reviewer 2 Report
Comments and Suggestions for Authors
Among food-borne pathogens, Clostridium spp have a crucial role, based on their strength in the environment and consequent potentiality of diffusion. In Your review, main aspects of contamination and health risks are treated and explained, with a worthwhile result. I would suggest to slightly enlarge 2.1 Chapter, "Pathogenicity", both for C. botulinum, C. perfringens and C. difficile, by adding some more elements that could complete that chapter: some more details about mechanisms of action by bacterial toxins could better Your explanation; the whole level of the article could be enhanced. Similarly, Chapter n°. 4 "Other aspects [...] environment" could deserve a light revision, simply adding some more detail about environmental diffusion of Clostridium spp bacteria. Table n°. 1 appears very interesting and complete: I would suggest to change the first row:
- actual: "2013 EFSA Journal 2015; 13(1):399"
to be changed as follows:
- new (suggested): "2013 EFSA (ref. # 28)"
and so on, for each cell in the first row.
Comments on the Quality of English LanguageEnglish construction in the paper seems well organized: in my opinion, a further and deeper reading could better the whole level, with minor revision/check
